# The Impact of COVID-19 on Young People’s Mental Health in the UK: Key Insights from Social Media Using Online Ethnography

**DOI:** 10.3390/ijerph19010352

**Published:** 2021-12-30

**Authors:** Rachel Winter, Anna Lavis

**Affiliations:** Institute of Applied Health Research, University of Birmingham, Birmingham B15 2TT, UK; r.winter@bham.ac.uk

**Keywords:** COVID-19, young people, mental health, social media, qualitative research, ethnography

## Abstract

There is increasing evidence of the psychological impact of COVID-19 on various population groups, with concern particularly focused on young people’s mental health. However, few papers have engaged with the views of young people themselves. We present findings from a study into young people’s discussions on social media about the impact of COVID-19 on their mental health. Real-time, multi-platform online ethnography was used to collect social media posts by young people in the United Kingdom (UK), March 2020–March 2021, 1033 original posts and 13,860 associated comments were analysed thematically. Mental health difficulties that were described as arising from, or exacerbated by, school closures, lost opportunities or fraught family environments included depression, anxiety and suicidality. Yet, some also described improvements to their mental health, away from prior stressors, such as school. Young people also recounted anxiety at the ramifications of the virus on others. The complexities of the psychological impact of COVID-19 on young people, and how this impact is situated in their pre-existing social worlds, need recognising. Forging appropriate support necessitates looking beyond an individualised conceptualisation of young people’s mental health that sets this apart from broader societal concerns. Instead, both research and practice need to take a systemic approach, recognising young people’s societal belonging and social contexts.

## 1. Introduction

Throughout the pandemic there has been cross-sectoral concern about the short- and long-term psychological impact of COVID-19 and the societal measures put in place to curtail its spread. In the United Kingdom (UK) this included a series of local and national lockdowns between 2020 and 2021, as well as school closures, compulsory mask wearing, social distancing, isolation and quarantining. Whilst the psychological impact of the pandemic has been seen across society, COVID-19 has been argued to have particularly affected young people’s mental health [1,2]. Research has suggested that young people have suffered mental distress [1,3], depressive symptoms [4,5], anxiety [6] and suicidal thoughts [7]. It is predicted that “1.5 million children and young people will require mental health support as a direct impact of the pandemic during the next three to five years” [8]. Creating appropriate mental health support has, therefore, already been argued to be an integral part of addressing both the short- and long-term consequences of the pandemic [9].

In young people this mental health impact has been attributed to various causes, notably disrupted schooling, online teaching, and social isolation [10,11]. Yet, there is also emerging evidence globally that some young people’s mental health has improved during the pandemic. In an England-based study, 35% of year 13s reported feeling happier [12]. A study in New Zealand found that 17.5% of participants (aged 18–75+ years) with pre-existing mental health conditions reported improvements to their mental health during the pandemic [13].

These contrasting findings point to “the need to now examine what accounts for variability in individuals’ mental health experiences during the pandemic” [14]. Understanding this will have significant implications for how mental health professionals tailor their support to the needs of specific groups of young people. This demonstrates the importance of in-depth explorations of young people’s own perspectives. These are key to forging appropriate short- and long-term support that is shaped by the concerns, needs and priorities of young people themselves.

### 1.1. Listening to Young People’s Voices: The Value of Social Media

COVID-19 precipitated a widespread move to increased time online, whether for work or socialising. In the UK 97% of children aged 5–15 went online in 2020, an increase on 2019 [15]. It is known that social media can offer a space for people to turn in moments of distress and seek support for mental health issues [16,17]. This has been documented amongst young people [18] and also during COVID-19, such as in relation to suicidality [19]. Social media discussions therefore afford crucial insights into the impact of the pandemic on young people’s mental health.

Research has explored the link between frequent social media exposure during COVID-19 and anxiety, [20], with the risk to mental health posed by negative news about the virus noted [21]. Recent studies have also investigated social media discussions about mental health and the pandemic, finding that people online have expressed feelings such as loneliness [22] and fear [23]. However, to date, research into the insights gained from social media into mental health during the pandemic has used methods such as machine learning [22,24] as opposed to in-depth qualitative methodologies. In contrast, online ethnography offers a way to engage with the complexities of people’s experiences during the pandemic by analysing both the breadth and depth of online conversations. It can thereby provide a rich and nuanced understanding of young people’s perspectives on the pandemic and their mental health.

### 1.2. Aims

The aim of this research was to explore young people’s discussions on social media about the impact of COVID-19 on their mental health in the UK. We sought to investigate how young people described this impact online and analyse both their attributions of causality and their statements regarding support needs.

## 2. Materials and Method

This paper draws on two real-time ethnographic studies conducted across a range of social media platforms during COVID-19 (2020–2021). *Study One* explored UK-based online discussions of the pandemic, tracking across the diverse conversations that developed on social media as events in the UK unfurled. *Study Two* was a focused exploration of relationships between the pandemic and online self-harm and suicide discussions.

Throughout both these studies, we were able to listen to a range of voices across demographic groups and at multiple temporal moments. This facilitated an engagement with the similarities and divergences between the experiences and needs of different groups of young people and at various timepoints during the pandemic.

### 2.1. Data Collection

Underpinned by the interpretive qualitative framework of medical anthropology, which engages with the social, cultural and structural dimensions of a given phenomenon, online ethnography is a well-established method [25]. It is adaptable to textual and visual content, taking account of all forms of expression (including videos, GIFs, memes, emojis), and provides a holistic, comprehensive and contextual understanding of online discussions.

Across the two studies, data for this analysis were collected in 23 online spaces by both authors; these included online forums, *Reddit*, and *Twitter*. Online forums are spaces where people meet with each other and have, often in-depth, discussions; we focused on ones about mental health. Subreddits on *Reddit* are similar to these forums and comprise online communities with a shared interest in a given topic. *Twitter* posts, given their character limit, are shorter expressions. We collected 1033 original posts across these spaces, both textual and visual, and 13,860 associated comments between March 2020 and March 2021. To keep a contextual understanding of discussions, comments were never removed from the original post. Instead, threads of conversations were collected and analysed. Data collection was supplemented by ethnographic fieldnotes of observations and emerging analyses.

The scope of *Study One* was broad; our data collection responded to societal and political events around COVID-19 happening in real-time in the time in the UK, such as the instigation of lockdown or the mandating of mask wearing. Preliminary analysis of data showed that people were also contributing to discussions on such subjects by talking about the relationship between the pandemic and their mental health. *Study Two* built on these preliminary findings by conducting focused research into the relationship between COVID-19 and online self-harm and suicide discussions.

In response to emerging data on young people’s mental health in our *Study One* dataset, we systematically collated the key terms that were being used by young people in the conversations we had collected so far, both to describe how they were feeling and to denote causality. These first terms included, for example, ‘COVID’, ‘virus’, ‘lockdown’, ‘mental health’, ‘anxiety’ and ‘depression’, as well as broader, non-clinical terms, such as ’stressed’ and ‘lonely’. We then also collated the key terms related to the pandemic that was being linked to self-harm or suicidal feelings by young people in our *Study Two* dataset. This first combined list of key terms was then used to conduct focused data collection and analysis on the topic of young people’s mental health and COVID-19, which was situated within each of the two studies.

The first step was to re-input these search terms into social media (across the online forums, *Reddit* and *Twitter*). At this step and all subsequent ones in the process of data collection and analysis, only posts in which young people spoke about their mental health and explicitly linked this to COVID-19 were included. The inclusion criteria for posts were: written by young people; in English; centred on mental health; included mention of COVID-19 or COVID-19 restrictions; based in the UK.

The second step was to analyse this first round of focused data. This included searching the data for further key terms being used by young people to describe their mental health and attribute causality. For instance, a post originally found using ‘lock down’ and ‘depression’ might also discuss ‘online school’ or vice versa. This would then lead us to incorporate the new terms into ensuing searches. This ensured that, at each step, we were following the language used by young people themselves to describe their experiences. This also involved noting any colloquialisms or community-specific languages, such as misspelling of clinical terminology or mental health related terms. These too were then re-inputted.

Having found original posts, at each stage of data collection we also collected and analysed the associated comments. Comment sections provide a space where people can gather to share and compare their experiences and these, therefore, offer insights into a range of views and, often, differences and similarities among these. The comments on a post do not always include the same keywords as the original post as they may express a perspective on the same topic but in a very different way. As such, it is important to note that the comments would have been missed out of the dataset had we relied solely on gathering posts which only included the search terms. Therefore, both analysing the comments, and also the iterative process of data collection through which they were found, were crucial to ensuring that the study was responsive to the emphases and terminology of young people’s conversations. At each step, using the analysis of posts and comments to direct the next round of searches for posts ensured that any assumptions on our part that may have otherwise been built into the search terms were constantly challenged and overcome. This is particularly important to note in relation to diagnostic terms.

That the search terms used for the first round of focused data collection included clinical, or even diagnostic, terms, such as “anxiety” as well as lay terms, such as “stressed” ensued directly from our first round of data analysis across the two studies, noted above. They illustrate the explicit links that young people themselves were making on social media between COVID-19 and clinical ways of framing their mental health experiences. However, we also ensured, through the analysis of comments and the iterative data collection process noted above, that we did not only search for posts that used such terms. Thus, in mirroring young people’s use of such diagnostic language to frame their experiences, and also interrogating the context to this, in this paper we are following anthropologist Rebecca Lester in treading between “critical cultural analysis and descriptive consistency” [26]; we are acknowledging psychiatric categories themselves as cultural whilst also recognising how they are mobilised as “a common set of understandings” [26] through which young people articulate distress on social media. In addition, maintaining a focus on this clinical terminology allows this paper to complement and extend the emerging evidence of this explicit linkage between mental health experiences and COVID-19 in the wider literature. In so doing, our findings contribute to the growing body of evidence regarding what young people will need in order to support their mental health recovery post-pandemic.

### 2.2. Identifying Posts by Young People in the UK

Throughout the data collection steps noted above, we used both linguistic markers and the content of posts to isolate those written by UK residents, for example, those mentioning UK locations or discussing how a UK policy was impacting them. This was enhanced throughout by our explicit focus in both studies on following events in the UK so that conversations were mapped against events such as changes in guidance or the instigation or release of lockdowns.

In addition, various strategies were employed to identify and isolate posts written by young people (using the World Health Organisation’s definition of individuals aged between 10 and 24 years old). In the UK, 42% of 5- to 12-year-olds were found to use social media in 2021 [15]. Yet, it can be difficult to assess the age of social media users, unless they explicitly state this. Additionally, although social media platforms widely set age limits, it is known that children contravene the rules to have accounts. Therefore, we could not use these rules as a basis for assessing if someone was of the age that a social media platform permits. To combat this, we followed a number of measures to isolate posts that indicated that they were written by a young person. First, age was indicated through the content of posts, such as a person mentioning their age, being at school/university, homework, or living with parents/carers. Secondly, as we have done previously in relation to online self-harm content [18], we analysed the language of posts, using Schwartz’s ‘language of age’ [27] which indicates the differences in the language used by different age groups. Thirdly, data were collected across social media but also in online communities created by and for young people.

### 2.3. Analysis

Whilst the majority of data collected were textual, the dataset also included memes, photographs, emojis and gifs, which were used in both posts and comments. We included all these forms of expression within our data analysis. Emojis and gifs, for instance, can aid understanding of the emotional tone of a post and so their inclusion helped to contextualise posts and the meanings behind them. Both the multi-media nature of the data and the iterative relationship between data collection and analysis noted above required an adaptable analytic technique. Reflexive thematic analysis [28] was therefore selected as this can be applied to various forms of media. In each round of analysis we generated codes, which were then grouped into themes. In each further round of data collection and analysis these codes and themes were reflected on, rejected, or added to.

To increase the validity of the findings and reduce researcher bias, both researchers individually analysed the data and met regularly to discuss findings. Reflective diaries were also kept, in which both authors were able to consider any personal biases they may have had with regards to data collection and analysis, before then discussing these as a team. Throughout data collection we also continually analysed the ethnographic notes collected alongside the posts and comments that were gathered.

### 2.4. Ethics

Both *Study One* and *Study Two* received ethical approval from the University of Birmingham’s Science, Technology and Engineering Ethics Committee.

There are continual debates about what ‘public’ and ‘private’ mean in relation to social media data [29,30]. Public data are often defined as posts that do not require a login to view content. To adhere to only using public data, this research used *Twitter*, *Reddit* and forums where a log-in was not required. However, in addition to posts being public, it is vital to respect the anonymity, traceability and confidentiality of their authors. Therefore, there are no direct quotations or usernames in this paper. Any quotations presented are either common phrases that cannot be attributed or traceable to a single person or they are an amalgamation of more than one similar quotation. In presenting the findings in this way, we have not altered the terminology that young people themselves used to describe their mental health or attribute causality. In addition, this paper does not mention any specific search terms, other than the example keywords noted above in the overview of data collection, or any hashtags. These strategies for reporting data ethically were developed in consultation with an Advisory Group of young people with lived experiences of mental ill-health or distress.

## 3. Results

This section presents four key themes: the virus: loss, fear and other people; isolation and life inside the home; education at a distance; and thinking about the future.

### 3.1. The Virus: Loss, Fear and Other People

In relation to the virus itself, young people primarily spoke of fear and being “anxious”. Across online spaces, statements, such as these elucidated this:“I’m scared about getting COVID”“I’m scared because I’ve tested positive for COVID”“I got COVID and now I’m panicking”

However, this anxiety was less pronounced than that evidenced by young people’s discussions of seeing or hearing about how unwell others were. Many wrote about loved ones who had underlying health conditions for example, or who had already caught COVID-19, and they spoke of their fear of relatives becoming severely ill or “getting worse”. Across conversations, thus, young people described their anxiety at the threat posed to others’ health by COVID-19:“I keep getting anxious about my family and friends getting COVID”.

Other young people described having experienced the loss of a loved one due to COVID-19, and explicitly linked both depression and suicidal ideation to this bereavement. Of particular note were accounts of loss at a distance, with descriptions of being “depressed” at having been unable to say goodbye to a dying relative due to COVID-19 restrictions. Our ethnographic fieldnotes from July 2020 noted that: “the grief is starkly apparent when we read stories related to not being able to see a loved one when they were in hospital either dying from COVID or another illness.”

Frequent in conversations about the anxiety provoked by the virus itself were accounts of the fear of being the person who might transmit the virus to vulnerable relatives:“What if I give COVID to my (elderly family member) and they die from it?”

Many responded to this fear of being a threat by writing, “I wear a mask for/because…” noting vulnerable relatives or friends.

In the UK this fear was heightened by the reopening of schools post-lockdown when secondary school students turned to social media to write about being “stressed” and “crying all the time” because they felt schools were opening “too soon”. They wrote for example:“Schools aren’t safe”.

These narratives were replete with fear at the anticipation of returning to school. But, tracking these discussions past school reopening also demonstrated how for some young people these fears grew once in school and how, again, they related to the health of others. Statements, such as “people in my class/school have COVID”, were followed by “I’ve been near (family member/friend) I’m worried I might give it them”.

“Teachers and other students have COVID”.“People don’t wear masks correctly”.

Young people, therefore, spoke about ensuring that they followed government guidance to reduce the spread of the virus and described how this alleviated some of their “anxiety” or fear. Alongside these conversations were those in which young people spoke about tensions between themselves and their family or friends, when the latter groups were not, for instance, wearing masks. They described feeling frustration towards those not working towards reducing infection spread.

Across all these discussions, young people demonstrated both their affective responses to the virus itself, most notably fear and anxiety, and also how these were profoundly shaped by altruism and a situated concern for others; young people’s mental health during the pandemic was not only linked to concerns for their own health and wellbeing but embedded in their existing social worlds and networks.

### 3.2. Isolation and Life Inside the Home

Lockdowns meant that the lives of UK young people moved into the home, causing physical isolation from friends. Conversations on social media throughout 2020 and into 2021 demonstrated the impact of this through frequent statements, such as “I miss my friends”. Young people described struggling to adapt to virtual socialising, a lack of face-to-face interaction and also the loss of friendships, with some writing that the loneliness they were experiencing was causing them to feel “depressed” and “suicidal”.

However, conversations also revealed stark differences in coping with isolation and these were often explicitly linked by young people to their home environment, family relationships and/or level of support. Posts described living in what some called “toxic environments” and cited tensions between themselves and their families:“I am now stuck with them (my family) with nowhere to go”

This sense of entrapment was clearly heightened when young people lived in fraught or abusive family environments, which were explicitly linked in these discussions to depression, anxiety, self-harm and suicidal thoughts.

Ethnographic fieldnotes, May 2020: “Young people talk to each other about how they are “stuck” with the people who made them suicidal from before the pandemic. Isolation is causing increased exposure to the verbal abusive and aggressive interactions they experience at home.”

These findings highlight that even early on in the pandemic young people were describing being “suicidal” due to being “trapped” and “stuck”. Such conversations were frequent into 2021 and intensified in each period of lockdown. Suicidality was also described as exacerbated by the closure of spaces to which young people had previously “escaped”, such as with friends.

Against this background of difficult or dangerous home environments and/or increased isolation, many young people described experiencing a lack of support for their mental health during lockdown. Those whose mental health difficulties had preceded the pandemic recounted how access to school or community mental health services had decreased during isolation. They wrote of finding services overwhelmed when they tried to reach out. Many of those describing pre-existing mental health difficulties explicitly related reduced support with severe depressive episodes and suicidal thoughts, saying that they felt “disappointed”, “abandoned” or “had nowhere to turn”, all of which then exacerbated feelings of isolation, in a looping effect.

### 3.3. Education at a Distance: School and University Online

In the UK, schools moved between online and offline learning, apart from vulnerable children and those of keyworkers. Throughout the pandemic, social media was replete with statements, such as “I hate online school”. But, against that wider background, young people who described having previously had mental health challenges, such as depression or anxiety, recounted going “back” to these after having previously recovered, and attributing this to online schooling.

“I’ve not been this suicidal for years”“I’ve relapsed” (referring to self-harm)

In turn, young people without a history of mental illness echoed these statements, describing how online education had caused them to experience “depression”, “anxiety” and “thoughts of suicide” for the first time.

Young people, with and without pre-existing mental health conditions, described finding it challenging to engage with online lessons and self-directed study and yet widely felt that:“School is expecting the same level of work despite COVID”.

Across conversations, school-related mental health difficulties were often focused on school performance and perceptions of “failing”. Many young people drew an explicit link between their grades and mental health, with poorer mental health and falling grades described as going hand in hand, with bi-directional causality. Many wrote that they had “no more motivation or energy” for studying and learning, which led to grades lowering and a trajectory from a “grade A star” student to “barely passing”. As such, the challenges posed by online schooling led many young people to turn to social media to describe “losing their grip” on both their school grades and mental health in tandem.

For those young people with pre-existing mental health difficulties, some wrote that their “previously controlled depression and anxiety” had now returned because of this sense of failure. Struggling to pass assignments was described as causing them to feel “suicidal” or had led to self-harm relapses. Such affective challenges contributed to lowered motivation to work and engage with online teaching, and therefore, lower grades. Across both groups of young people, this cycle was described from the initial months of lockdown but worsened as the pandemic continued.

As noted above in relation to isolation, some young people described being “lonely” due to school closures. Yet, others felt their mental health had benefitted specifically from these:“I don’t know about others but my suicidal thoughts have improved being away from school”.

This was not an isolated experience, with conversations across platforms noting an improvement in mental health. Such posts were usually from young people who described a history of depression, anxiety, self-harm and/or suicide. They wrote about the advantages of being away from a space where they were “bullied” or surrounded by “cliques”. Therefore, they expressed fear at schools reopening.

However, there was a further way in which school closures impacted mental health; they meant that young people missed experiences they had been looking forward to or working towards.

Ethnographic fieldnotes, August 2020: “‘COVID stole prom/graduation away from me’: In the comments it’s clear that this is a shared experience, with people feeling that COVID had stolen different moments from each individual, the result, they write, is feeling depressed or suicidal, as something which they were looking forward to has been taken away.

### 3.4. Thinking about the Future

In addition to articulating the loss of events that had been imminent, such as proms, young people used social media to write of their hopelessness about the longer-term future. Across conversations this was linked with experiencing suicidal thoughts, depression and anxiety.

Many young people discussed how prior to the pandemic they had imagined their lives would be better than they were now or that they now could be in the future. Linked to expressions of suicidality in particular, young people’s discussions of how they had thought they would leave school or university and get a flat, move out of their parents’ house, or have a job, were replete with articulations of fear that these would all be less viable during and after the pandemic. This worry about the future was something that many young people described continually thinking about.

This sense of hopelessness, fear and “feeling anxious” increased as the pandemic unfurled, with discussions of the social and economic repercussions of COVID-19 becoming more frequent across social media platforms. Articulations of fear about how the economic consequences of COVID-19 would curtail future job prospects were clearly associated with descriptions of feeling “depressed” and, again, “suicidal”. These employment fears were mentioned by both university students and those at school. The latter also feared that lower grades due to online schooling would impact:

Ethnographic fieldnotes, March 2021: “Across all the social media spaces we are exploring, young people are discussing the economy, predicting how many years it’ll be before it recovers. We’ve seen anything from a year to several decades. Secondary school children to graduates keep saying that there is no hope of ‘good’ jobs or working in the field they want because of a recession. ‘I might as well give up now’ is frequently echoed alongside thoughts of suicide.”

Threading through these discussions were young people’s descriptions of how they had witnessed parents losing their jobs during the pandemic or experiencing reduced family finances; both compounded the hopelessness felt about their own employment prospects.

## 4. Discussion

This paper has highlighted the myriad ways in which the pandemic has impacted young people’s mental health, which they frequently described in ways that echoed diagnostic nosology. By applying terms like “depression” and “anxiety” to their experiences, young people across social media made an explicit causal link between COVID-19 and poor mental health, citing experiences of depression, anxiety, self-harm and, crucially, suicidality. However, social media also comprised discussions of how changes to young people’s social environments resulting from the pandemic had improved their mental health. Therefore, our findings align with recent research that has argued that COVID-19 control measures have had no singular or universal impact on individuals’ mental health [31,32]; the impact has been complex, nuanced and contextual.

Throughout the pandemic, there have been concerns and growing evidence that school and university closures [10,33] and social isolation [11] have negatively impacted mental health. The effect of social isolation and loneliness on children and young people was a concern before the pandemic and has been heightened throughout. Both have been linked with depression, poor mental health, and anxiety [34,35,36]. This correlates with our findings that social isolation ensued from being away from friends and was potentially compounded by fraught or abusive family environments. Young people described feeling trapped, depressed, anxious, and suicidal.

Disrupted education, in terms of both schools and universities, has also been cited as a cause of poorer mental health during the pandemic [37,38]. Throughout, UK public and political discourses have widely framed school in particular as a safe space to which children should return to improve their mental wellbeing. On the one hand, our findings align with this in that some young people attributed their experiences of anxiety, depression and suicidality to online schooling and isolation. They linked these with poorer social relationships, missing friends, and lower grades. However, it is also crucial to recognise that there have been continual conversations amongst young people on social media about how being away from school improved their mental health and wellbeing, with descriptions of “anxiety returning” when schools and universities reopened. As Scott, McGowan and Visram [39] have also suggested, “it may be that schools are not the panacea to social ills”. COVID-19 has highlighted the different experiences that young people have at school, it being both a space of socialisation and safety, but also one of anxiety.

Social media conversations have also highlighted the urgent need to recognise that some young people fear catching COVID-19 from school or university and that the removal of infection control measures, such as masks caused some anxiety and fear. Our data indicated that young people have had differing levels of “health anxiety”, [40] during the pandemic. In our dataset, moreover, such anxiety was for the health of others as much as one’s own. It is currently unknown how long the COVID-19 pandemic may last or the lasting impact that health anxiety may have on individuals. It will therefore be important to provide resources and support tailored for those experiencing health anxiety [40]. As young people return to educational institutions, it is also vital to understand these mixed emotions towards school and university and reflect on how to ensure that schools and universities do offer safety, both from the virus and more widely.

Young people have also experienced numerous forms of grief and loss due to COVID-19. With regards to bereavement, the dataset evidenced the combined grief of both losing someone and not being able to say goodbye due to restrictions in hospitals. Bereavement in adolescence is a known risk factor for depression [41], with parental bereavement linked with lower educational aspirations and competence in work [42]. In addition, it has been suggested that grief experienced during COVID-19 has associated “feelings of guilt, shame, isolation and abandonment” [8] and some young people described this grief as linked with depression and suicidal thoughts. Given the potential “grief pandemic” [43] that society may face, it is evident that this complex grief necessitates reflecting on in order to support young people.

Added to this complex grief is the collective grief that resonated through young people’s descriptions of how COVID-19 has “stolen” important events, such as graduation or prom. This sense of mourning the loss of life events tallies with other research [44] and was seen in our study to lead to decreased motivation to engage with school work, which was linked with depression and suicidal thoughts. Our findings also point to the need to recognise prospective grief and a sense of lost future opportunity. In so doing, our data align with other studies that have found that young people feel that COVID-19 will cause reduced job opportunities [45] and that they feel hopeless about the future [46].

Concern has been raised about the potential impact of COVID-19 on suicide rates. However, to date in the UK, there is no evidence to suggest a rise in the number of completed suicides [47]. There is, however, some suggestion that suicidal thoughts may have increased during this period [7]. Across our dataset, young people with a history of suicidal thoughts described how these had increased, and others recounted that they were experiencing these for the first time.

Across platforms and conversations, depression, anxiety and suicidal thoughts were described in ways that showed them to be new experiences provoked by the pandemic for some young people, and also as previous difficulties that had recently returned for others. Whilst there were clear themes around causality, such as related to schooling, fear of giving the virus to others, and the long-term economic impact of COVID-19, the way in which each of these impacted a particular young person depended on their social context and support networks. For instance, poorer educational attainment through lowered grades during the pandemic caused some to feel anxious but others to feel suicidal. As such, online conversations about COVID-19 and mental health show the messy and entangled nature of causality and illustrate the multiple and cumulative causes of anxiety, depression, suicidal thoughts and self-harm. This finding aligns with Scott, McGowen and Visram [39] diary-based study in which young people spoke about the differing impacts that lockdowns and school closures were having on their mental health. As such, our findings demonstrate that young people’s mental health through the pandemic has been profoundly shaped by their social contexts and societal belonging.

Crucially, young people’s narratives on social media have shown their mental health to be profoundly connected to the wellbeing of others. Many described experiencing depression at isolation, for example, whilst also understanding and supporting the need for virus control measures. In turn, whilst some discussed concerns about personally catching COVID-19, our data support Widnall et al.’s finding [48] that a significant concern for young people was their family and friends becoming unwell. Their articulations of fear at the economic repercussions of COVID-19 were also focused on wider society, especially when seeing family or friends losing jobs, as well as their own prospects. As such, individual affective challenges were interwoven with and underpinned by a broader sense of societal belonging and responsibility. This shows the need to move beyond a pervasive “psychocentrism” [49] in many current public and political discourses about young people’s mental health during the pandemic. A tendency to hold up singular causes, such as school closures, has pinpointed solutions in disconnected acts such as the reopening of schools or allowing the “young” or the “healthy” freedom. In contrast to the sense of “embodied belonging” [50] that runs through young people’s narratives on social media, such solutions risk doing more harm than good through a politicised and enforced unbelonging that fails to recognise the social embeddedness of young people’s mental health.

## 5. Limitations

Whilst social media offers crucial insights into the lifeworlds and experiences of young people, we recognise that not all use social media and, of those that do, only some will be doing so to discuss their mental health experiences. This paper has, therefore, only engaged with the voices of those who have written about these topics online. In addition, this paper has been less able to present the narratives of those young people whose mental health was not affected during the pandemic.

Due to our ethical approval and protocol, we only collected data from platforms that did not require a log-in to access posts. This means that there may be conversations in closed or private spaces which do not align with those that were analysed in this study.

Part of the first round of data collection included (but was not limited to) searches for conversations that mentioned mental health specifically in diagnostic terms, as directed by the search terms identified through our prior preliminary analysis. As such, this round of data collection risked missing out on conversations amongst young people who did not describe their experiences in this way. However, analysis of that first round iteratively directed the next round of search terms and data collection, which challenged or contextualized the earlier data that were collected using more clinical terminology. Furthermore, by analysing the comments accompanying all the collected posts we were able to explore the diverging ways in which mental health experiences were expressed, with comments often differing from the original post.

In social media data collection, it can be challenging to determine the age of individuals. Not everyone writes their age, and demographics are not always reliable on social media as young people sometimes lie about their age to gain access to platforms. Therefore, there is a small chance that data not written by young people have been included in the dataset. For instance, in relation to posts about university or describing being in their first year, we acknowledge that these may have been written by mature students. However, due to the comprehensive and contextual nature of our data collection, with comments as well as posts included in the analysis, we do not envisage that this small potential has jeopardized the findings.

## 6. Conclusions

This paper has highlighted the range of ways that COVID-19 has impacted young people’s mental health. Difficulties that were described as arising from, or exacerbated by, school closures, lost opportunities or fraught family environments included depression, anxiety and suicidality. Young people also recounted anxiety at the virus’s impact on others. Yet, some described improvements to their mental health, away from prior stressors, such as school. It is evident that, in moving beyond the pandemic, mental health services will need to take account of the many diverse experiences of young people. Support may be required not only for those whose mental health deteriorated during the pandemic but also for those who felt that being away from educational institutions improved their mental health and who were profoundly anxious about returning to school or university. For mental health services, thus, the study has further emphasised the need for “investment in online and face to face psychological counselling services” [51] post-pandemic. These are needed to account for the heterogeneous ways in which young people’s mental health has been affected, and also to look to the future to consider how the economic and societal consequences of the pandemic may further impact wellbeing [51].

Ensuring that priorities for recovery are driven by the needs of young people themselves necessitates recognising that solutions lie in complex and connected reflections on causation and context. Forging appropriate support necessitates looking beyond an individualised conceptualisation of young people’s mental health that sets this apart from broader societal concerns. As we move through, and out of the pandemic, thus, supporting young people will require whole systems thinking and, as Holmes et al. [52] have suggested, interdisciplinary research; both the social sciences and humanities, alongside the clinical sciences, will be key to ensuring recovery at both an individual and societal level.

## Data Availability

Data have been anonymised to protect the traceability of the posts used in this study. Data cannot currently be made publicly available as this would potentially make social media participants identifiable.

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
