# Peer review of "The Impact of COVID-19 on Young People’s Mental Health in the UK: Key Insights from Social Media Using Online Ethnography"

_ijerph, 2021, doi:10.3390/ijerph19010352_

Round 1

Reviewer 1 Report

Dear authors!
I found your article an interesting ethnographic research devoted to the narrative analysis of online communication of young people. This study is devoted to an urgent topic - the reaction of society to the COVID-19-pandemic. The results may be interesting to public health professionals, psychologists and psychiatrists.
In order to increase the degree of persuasiveness of the results obtained, I consider that it is important to answer for following questions in the description of the data collection methodology: Was randomization is used for the representativeness of the results? Why did you choose decrypted websites and social networks? Which social networks are the most popular among young people in your country?

I hope that my comments will help improve the article and get final approval for publication.

Author Response

Many thanks for your helpful review. Please see the attached file for detailed responses. 

Author Response

(The authors gave the same response as above.)

Reviewer 3 Report

This is a very interesting paper on a timely issue, which has been neglected during the first phases of the pandemic. Therefore, I read with extreme interest this paper and found it very relevant. 

However, I have some minor suggestions to further improve the quality of the paper:

  • in the Introduction, authors should cite some relevant papers for explaining the impact of the pandemic on mental health and its relevance for psychiatry (e.g., please see Unützer J, Kimmel RJ, Snowden M. Psychiatry in the age of COVID-19. World Psychiatry. 2020 Jun;19(2):130-131; Adhanom Ghebreyesus T. Addressing mental health needs: an integral part of COVID-19 response. World Psychiatry. 2020 Jun;19(2):129-130; Fiorillo A, Gorwood P. The consequences of the COVID-19 pandemic on mental health and implications for clinical practice. Eur Psychiatry. 2020 Apr 1;63(1):e32; Marazziti D, Stahl SM. The relevance of COVID-19 pandemic to psychiatry. World Psychiatry. 2020 Jun;19(2):261).

2. It would be useful to add a table or a figure in order to summarize the main themes identified by the qualitative analyses. Moreover, have you had the opportunity to collect also socio-demographic information such as age, gender, level of education, etc? If not, this should be acknowledged among study's limitations.

3. Authors should discuss the issue of "COVID- health anxiety" as described by Tyrer (Tyrer P. COVID-19 health anxiety. World Psychiatry. 2020 Oct;19(3):307-308. doi: 10.1002/wps.20798. PMID: 32931105; PMCID: PMC7491644.).

4. The problem of social isolation and loneliness is particularly relevant for young people, please discuss the clinical and practical implications of these aspects, even considering the phenomenon of hikikomori (several papers have been published in this issue).

5. in the final part of the discussion, authors should clearly report the practical implications of their study. In particular, you should discuss these implications considering recent position papers issued by international associations such as Stewart DE, Appelbaum PS. COVID-19 and psychiatrists' responsibilities: a WPA position paper. World Psychiatry. 2020 Oct;19(3):406-407; Wasserman D, Iosue M, Wuestefeld A, Carli V. Adaptation of evidence-based suicide prevention strategies during and after the COVID-19 pandemic. World Psychiatry. 2020 Oct;19(3):294-306; McDaid D. Viewpoint: Investing in strategies to support mental health recovery from the COVID-19 pandemic. Eur Psychiatry. 2021 Apr 26;64(1):e32; Kuzman MR, Curkovic M, Wasserman D. Principles of mental health care during the COVID-19 pandemic. Eur Psychiatry. 2020 May 20;63(1):e45).

Author Response

(The authors gave the same response as above.)

Reviewer 4 Report

This is an important and interesting study. The manuscript is very well written and this topic is of interest to a wide audience.

I have only a few suggestions for improvement:

I would implement the introduction part and the aims of the study. Several studies conducted have covered similar topics (e.g., Giannini A.M. et al.), but not with this type of analysis, it would be helpful to add the importance of this study.

Typo line 124

I would better explain the sites where the analysis has been done (e.g. facebook, myspace, Instagram...?)

Among the limitations mention the fact that not all data were traceable, but only those that could be found with public profiles.

It is worth including the practical applicability of the research in the conclusions.

Best Regards

Author Response

(The authors gave the same response as above.)

Round 2

Reviewer 2 Report

Dear authors,

Thank you for your careful and thorough response to the comments. The manuscript has been greatly improved by the elaborations to the Methods and Limitations sections. The addition of the Conclusions section is very helpful for the reader. 

Two minor points:
a) The 'Conclusion' heading should be 'Conclusions'.
b) The Conclusions heading should be placed below the Limitations section - as per the IJERPH author guidelines. 

Reviewer 4 Report

ok